# Sequential Extraction of Bioactive Saponins from *Cucumaria frondosa* Viscera: Supercritical CO_2_–Ethanol Synergy for Enhanced Yields and Antioxidant Performance

**DOI:** 10.3390/md23070272

**Published:** 2025-06-28

**Authors:** Jianan Lin, Guangling Jiao, Azadeh Kermanshahi-pour

**Affiliations:** 1Biorefining and Remediation Laboratory, Department of Process Engineering and Applied Science, Dalhousie University, Halifax, NS B3J 1B6, Canada; jianan.lin@dal.ca; 2AKSO Marine Biotech Inc., Hacketts Cove, NS B3Z 3K7, Canada

**Keywords:** saponins, *Cucumaria frondosa*, by-products, biomass valorization, green extraction, scCO_2_ technique

## Abstract

This study investigates the sequential extraction of lipids and saponins from *C. frondosa* viscera. Lipids were extracted using supercritical carbon dioxide (scCO_2_) in the presence of ethanol (EtOH) as a co-solvent. Subsequently, the lipid-extracted viscera underwent three saponin extraction approaches, scCO_2_-scCO_2_, scCO_2_-EtOH, and scCO_2_-hot water, resulting in saponin-rich extracts. Process parameter investigation for saponin extraction from scCO_2_-defatted viscera revealed minimal effects of temperature, pressure, extraction time, static extraction, and EtOH concentration on saponin yields, allowing for milder operational conditions (35 °C, 20 MPa, 30 min dynamic extraction, 75% EtOH at 0.5 mL/min) to achieve energy-efficient recovery. Continuous EtOH feeding predominates the scCO_2_ extraction of saponins. The sequential scCO_2_ extraction of lipid and saponins yielded saponins at 9.13 mg OAE/g, while scCO_2_ extraction of lipid followed by a 24 h 70% EtOH extraction of saponins achieved 16.26 mg OAE/g, closely matching the optimized ultrasonic-assisted extraction of saponins (17.31 mg OAE/g) from hexane-defatted samples. Antioxidant activities of saponin-rich extracts obtained in the sequential scCO_2_-EtOH extraction (17.12 ± 4.20% DPPH scavenging) and the sequential scCO_2_-scCO_2_ extraction (16.14 ± 1.98%) were comparable to BHT (20.39 ± 0.68%), surpassing that of hexane-defatted ultrasonic extracts (8.11 ± 1.16%). The optimized scCO_2_-EtOH method offers a sustainable alternative, eliminating toxic solvents while maintaining high saponin yields and bioactivity.

## 1. Introduction

Saponins are high-molecular-weight glycosides composed of diverse sugar moieties attached to triterpene or steroid aglycones [1]. While they are predominantly found in higher plants as defence compounds against pathogens and herbivores [2], increasing evidence suggests their presence in lower marine animals, particularly in the marine phylum *Echinodermata*, including the classes *Holothuroidea* (sea cucumbers) and *Asteroidea* (starfishes) [3]. As research into their pharmacological and biological properties develops, saponins have been identified as bioactive ingredients of traditional Oriental medicines and pharmaceutical drugs, exhibiting diverse biological activities such as immunostimulatory properties, as well as anticancer, antimicrobial, antifungal, anti-oxidative, anti-inflammatory, and antiviral activities [1,4]. Additionally, they have been shown to combat obesity, improve cardiovascular health, lower hypertension, and reduce diabetes risk [5,6,7,8,9,10].

Recent studies have identified sea cucumbers as a particularly rich source of triterpene glycosides, which function both as chemical defences against predators and as bioactive compounds with remarkable pharmaceutical potential [4,6,7,10,11,12]. Among their diverse biological activities, sea cucumber-derived saponins have been reported to exhibit antioxidant potential. Studies have shown that saponins isolated from various sea cucumber species possess radical-scavenging and reducing activities, contributing to their therapeutic relevance [13]. To date, approximately 300 triterpene glycosides have been identified across various sea cucumber species. The saponins extracted from *C. frondosa* include a diverse range of triterpene glycosides, with frondoside A being the most well-documented [14]. Other identified saponins include frondosides B, C, D, E, F, isofrondoside C, frondoside A derivatives, and the dimeric pentasaccharide frondecaside [11,14,15,16]. A recent study identified six previously undetected saponins and two unknown compounds in *C. frondosa* extracts [17].

While the body wall is commonly studied as the main source of sea cucumber saponins [18,19], the viscera, often discarded during processing, have emerged as a promising and underutilized alternative. Rich in saponins, as well as lipids and polysaccharides, the viscera offer potential for integrated biorefinery applications [20,21]. Their use not only adds value to processing by-products but also supports sustainability goals by reducing waste. The high saponin content and ready availability of viscera make them an economically and environmentally attractive raw material for extraction.

Developing an efficient method for extracting and quantifying saponins from marine by-products is crucial for maximizing their utilization. Extraction efficiency is influenced by factors such as biomass feedstock properties (e.g., moisture content, particle size), solvent type, and extraction conditions (e.g., temperature, solvent-to-feedstock ratio). Biomass pretreatment, including drying, particle size reduction, and defatting with lipophilic solvents like ethyl acetate or hexane, is often required to improve solvent penetration and enhance mass transfer during extraction [22,23]. Conventional saponin extraction on marine biomass typically involves solvents such as methanol (MeOH), ethanol (EtOH), water, or aqueous alcohol under mild heating [24]. The extracts are partitioned with n-butanol, and the pooled fractions are concentrated to obtain crude saponins for analysis [23,25]. However, traditional extraction methods have drawbacks, including prolonged extraction times, high solvent consumption, and limited efficiency. Consequently, researchers are exploring greener and more efficient alternatives, such as ultrasonic-assisted extraction [17], with supercritical carbon dioxide (scCO_2_) extraction emerging as a promising approach.

Supercritical fluid extraction is an efficient method for valorizing food by-products, utilizing supercritical fluids that combine the permeation ability of gases with the solvency of liquids. ScCO_2_ is commonly used due to its non-toxic, non-flammable, cost-effective, and eco-friendly properties [21]. Although the scCO_2_ extraction technique has been extensively investigated for plant-derived saponins, including those from tea camellia seed cake [26], quinoa bran [27], *spina Gleditsiae* [28], Brazilian ginseng [2], and sisal hemp [29], with optimal conditions falling in ~45–60 °C and ~30–50 MPa using EtOH as co-solvent, its application to marine-derived saponins remains underexplored. The limited research available has primarily focused on the scCO_2_ extraction of saponins from *C. frondosa* body wall, reporting a yield of 1.1 g triterpene glycosides per 60 g of freeze-dried samples [30]. Since marine-derived saponins often coexist with lipids and other non-polar compounds [11], the scCO_2_ extraction technique is regarded as a promising alternative, offering tunable selectivity, mild extraction conditions, and reduced reliance on organic solvents. It also enables efficient sequential extraction of both lipophilic and hydrophilic components.

Sequential extraction techniques have emerged as a practical approach for the sustainable valorization of food by-products through multiple extraction steps. ScCO_2_ extraction is commonly used as the first step, helping to extract lipids [31]. Once the lipids are removed, the remaining defatted biomass can be processed further with techniques like pressurized liquid extraction [32,33], ultrasound-assisted extraction [33], enzyme-assisted extraction [34], or subcritical water hydrolysis [35] to recover other valuable bioactive compounds, such as phenolics and bio-peptides. This approach not only increases the efficiency of extraction but also supports the sustainable use of food waste by maximizing the recovery of different compounds from the same raw material. However, the sequential valorization of sea cucumber processing by-products has yet to be thoroughly explored in the literature.

Liquid chromatography–mass spectrometry and similar chromatographic techniques are commonly used to analyze sea cucumber saponin fractions, such as frondoside A and cladoloside A, with reported concentrations ranging from 0.01 to 0.15 mg/g in dried samples (0.05 to 0.73 mg/g in dried extracts) [17,36]. While these methods offer accurate analysis, multiple saponin fractions with varying structures and chromatographic behaviours, along with some saponins being present at low concentrations below detection limits, complicate quantification. This requires specialized knowledge, complex sample preparation, and optimized chromatographic conditions. Additionally, the high cost and time-intensive nature of the process limit their application. Spectrophotometric quantification of total saponins is becoming popular for its simplicity and cost-effectiveness. It relies on colorimetric reactions, where saponins turn lavender to purple upon reacting with vanillin–acetic acid–perchloric acid reagents. The resulting chromophore is measured with an ultraviolet–visible (UV-Vis) spectrophotometer, and saponin concentrations are determined through standard calibration [37,38]. However, spectrophotometric saponin quantification faces challenges such as matrix interferences from lipids [39]. Thus, there is a clear gap in understanding the importance of sample pretreatment and matrix correction to improve the accuracy of saponin quantification.

This study explored the subsequent extraction of saponins from seafood processing wastes, *C. frondosa* viscera, using scCO_2_, EtOH, and hot water following scCO_2_ lipid extraction. The process variables influencing the sequential scCO_2_ extraction of sea cucumber saponins, including temperature, pressure, co-solvent EtOH concentration, and extraction time, were examined using a factorial and screening design for the first time. Additionally, the study assessed whether saponins were co-extracted during scCO_2_ lipid extraction and evaluated the feasibility of subsequent scCO_2_ extraction for saponin recovery from lipid-extracted viscera. The impact of interference compounds (e.g., *lipids*) in the viscera matrix on saponin extraction was further examined. Yields and bioactivities of saponins obtained via various extraction techniques were compared to determine the most efficient and viable extraction method. Through this systematic investigation, the study seeks to advance scalable and sustainable sequential extraction methodologies for marine by-product-derived bioactive compounds.

## 2. Results and Discussion

### 2.1. Impacts of Interference Compounds on Saponin Quantification and Extraction

The conventional EtOH extraction of native and defatted *C. frondosa* viscera samples without further purification resulted in significantly different analytical saponin yields, which is associated with the interference of compounds such as lipids (Table 1). Shang (2016) reported that glucose, glycine, and palmitic acid can react to form pigments exhibiting maximum UV absorbance at 540 nm [39]. Furthermore, a colorimetric sulfo-phospho-vanillin method was developed for rapid quantitative analysis of lipids, examining that similar reagents could cause colour reactions with lipids [40]. Since sugars, amino acids, and fatty acids are commonly co-present with or within saponins, these components may contribute to overestimating the total saponin contents. Shang (2016) observed that the contribution of these constituents varied between 1.82% and 8.57% in ginseng [39]. In the current study, the interference caused by nontarget compounds in saponin yield determination was up to 28.03% (i.e., compared yields resulting from native and hexane-defatted viscera using 100% or 70% EtOH-(38.22–10.19) mg oleanolic acid equivalent (OAE)/g = 28.03 mg OAE/g = 28.03%, (36.34–16.44) mg OAE/g = 19.9 mg OAE/g = 19.9%; Table 1). Recently, Le Bot et al. (2022) developed a colorimetric method for quantifying total triterpenoid and steroidal saponins in plant materials and also incorporated a dichloromethane pretreatment prior to saponin extraction [37].

Moreover, reflux extraction of native and hexane-defatted *C. frondosa* viscera, followed by purification, excluded the interference from undesired compounds but showed significant differences in the saponin yields (4.18 vs. 6.71 mg OAE/g; Table 1). The phenomenon suggests lipid extraction improves saponin yields by removing hydrophobic barriers, enhancing solvent penetration, and disrupting cellular structures that trap saponins.

Therefore, defatting and purification are key steps in improving saponin extraction from sea cucumber viscera by reducing lipid interference, enhancing solvent penetration, and boosting yields. Defatting removes non-polar lipids that block the extraction of saponins by polar solvents, while purification helps eliminate the overestimation caused by interfering compounds, ensuring more accurate quantification. For instance, while scCO_2_ extraction removes most lipids in the first step, some residual lipids may remain in the system. These residual lipids can then be carried over into the second collection vial during subsequent saponin extraction, leading to an inflated estimate of the saponin yields if not properly purified.

### 2.2. Sequential ScCO_2_ Extraction of Saponins

To systematically explore the conditions for subsequent saponin extraction using scCO_2_, a stepwise experimental approach was implemented. This included preliminary single-factor experiments (Appendix A) followed by a definitive screening design (Appendix A) and ultimately replicated single-factor validations (Appendix A), allowing for a comprehensive assessment of critical process variables.

Besides those variables that have been proven to impact the saponin extraction in the conventional procedure (temperature, pressure, extraction time, and EtOH concentration), static extraction time and co-solvent loading parameters were explored under a fixed condition: 55 °C, 35 MPa, 45 min of dynamic extraction, and 75% EtOH used as co-solvent. The one-single-factor experimental results recommended 10 min of static extraction time with an initial addition of 3.75 mL EtOH, followed by continuous feeding of EtOH at 0.5 mL/min.

Subsequently, a definitive screening design (resolution IV design) was implemented to examine the impacts and the ranges of proposed parameters, temperature (Temp, 35–75 °C), pressure (Pres, 20–50 MPa), dynamic extraction time (DET, 30–90 min), static extraction time (SET, 0–30 min), initial EtOH volume (Vol, 0–7.5 mL), and EtOH concentration (Conc, 50–100 vol%) (Appendix A). Based on the Pareto chart of standardized effects (Appendix A), none of the main effects, interaction terms, or quadratic terms had a statistically meaningful impact on total saponin yields (i.e., all bars on the chart were grey, further confirming that no terms were retained in the model during stepwise regression). The results suggest that within the studied parameter ranges, saponin extraction remains relatively unaffected by changes in temperature, pressure, extraction times, initial EtOH volume, or EtOH concentration.

To further validate the insignificance of the two main scCO_2_ system criteria and reaffirm the definitive screening design findings, single-factor experiments with varying temperatures (35 vs. 75 °C) and pressures (20 vs. 50 MPa) were conducted (Appendix A), which reinforced that these variables do not critically influence saponin recovery, with *p* = 0.0849 and 0.6945 > 0.05.

Moreover, saponins were co-extracted during the scCO_2_ lipid extraction, as these amphiphilic natural products can easily attach to lipids [41], facilitating their simultaneous recovery. The noticeable differences in saponin contents in the lipid-rich extracts were attributed to the uneven extracts that remained within the supercritical fluid system after the initial extraction. During subsequent extraction cycles, these residual compounds were gradually eluted, thereby influencing the overall saponin measurements in the second stage. The current study found (2.90 ± 1.33) and (2.59 ± 1.24) mg of OAE/g of samples on a dry weight basis in the lipid-rich extracts and the rinse (washing solutions from the empty system after lipid extraction), showing a dynamic distribution of saponins during the extraction process.

Noticeably, 70% EtOH extraction significantly enhances the saponin yields compared with 100% EtOH extraction (16.44 vs. 10.19 mg OAE/g; Table 1). The improvement can be attributed to the synergistic effects of water in the extraction process. As amphiphilic molecules with hydrophilic sugar moieties and hydrophobic aglycone structures [42], saponins exhibit higher solubility in aqueous EtOH [43]. Water in 70% EtOH improves the balance between polar and non-polar properties, promoting matrix penetration and facilitating saponin release and diffusion. In contrast, absolute EtOH lacks the polar water component, limiting the ability to dissolve the hydrophilic sugar chains in saponins and disrupt hydrogen bonding, resulting in lower extraction efficiency.

Although EtOH was used as a co-solvent in the scCO_2_ extraction system, varying its concentrations did not significantly affect saponin yield within the tested range. This outcome suggests that once a certain polarity threshold is reached, further changes in ethanol concentration offer limited benefit to saponin diffusion under supercritical conditions.

In practical terms, 75% EtOH proved more favourable than 50% EtOH as a co-solvent because of improved process performance during depressurization. This is attributed to the physicochemical properties of EtOH–water mixtures. The viscosity of the water/EtOH reaches a maximum at about 50% ethanol [44], and a higher EtOH concentration lowers the surface tension [45]. Additionally, 75% EtOH has a lower viscosity and a reduced surface tension, allowing CO_2_ to escape more readily during depressurization. As such, 75% EtOH was selected as the recommended co-solvent concentration.

The continuous addition of the EtOH co-solvent is the principal factor modulating scCO_2_ polarity and governing saponin elution from the marine biological matrix. Bitencourt et al. (2014) also reported higher yields and better surfactant activity of Brazilian ginseng saponins with the aid of continuous co-solvent draining (EtOH or a 70% hydroalcoholic solution) in scCO_2_ extraction, among which 70% EtOH yielded the majority of saponins, revealing that the synergistic use of EtOH with scCO_2_ is crucial for both yields and functionality of saponins [2]. Other process parameters appear secondary, offering little yield enhancement within the tested range, as indicated by the definitive screening design results. These findings suggest that the extraction of saponins using scCO_2_ is highly dependent on surpassing a certain polarity threshold, which is enabled by co-solvents, after which the marginal gains from increasing pressure, temperature, or extraction duration diminish significantly. A study supported this observation by finding that raising pressure from 20.7 to 48.3 MPa, with other conditions fixed at 110 °C and a modifier to solid ratio of 4:1, had no significant effect on the total ginsenoside yield from *Panax quinquefolius* [46]. However, the minimal effects of temperature changes may be attributed to a lack of a single supercritical operating phase. A higher concentration of co-solvent can significantly influence the critical temperature of the CO_2_-co-solvent system. For instance, when the EtOH mole fraction reaches 30%, the critical temperature of the CO_2_-EtOH mixture rises to around 104 °C, compared to 31 °C for pure CO_2_ [47]. As a result, the proposed conditions may fall outside a true single supercritical phase, reducing the sensitivity of the saponin extraction to temperature changes. Although higher temperatures (>100 °C) raise safety and sustainability concerns, future studies should investigate the phase behaviour of CO_2_-EtOH-water in supercritical systems and better clarify the roles of temperature and co-solvent interactions in saponin extraction.

The lack of significant differences across the tested ranges indicates that the process is highly tolerant to operational fluctuations, which is a desirable trait for large-scale industrial applications where the consistent performance of the scCO_2_ extraction technique is critical due to its inherent variations and the challenge of precise control over these factors. From a process engineering standpoint, these findings support the use of simplified, milder extraction conditions, such as 35 °C, 20 MPa, 30 min dynamic extraction, 10 min of static extraction with an initial addition of 3.75 mL EtOH, followed by continuous feeding of 75 vol% EtOH at 0.5 mL/min. While these conditions do not necessarily maximize recovery, they ensure comparable yields while enhancing energy efficiency by operating at lower thermal and pressure settings for a shorter duration. This streamlined approach enhances process scalability and demonstrates strong potential for sustainable industrial implementation.

### 2.3. Sequential Extraction: ScCO_2_ Extraction Followed by Conventional Extraction

To explore the alternative to sequential scCO_2_ extraction with higher saponin yields, scCO_2_ extraction followed by hot water or 70% EtOH extraction and the effect of extraction time were investigated (Table 2). The extraction time of 24 h was determined as the optimal extraction in both cases since the yields at 24 h extraction achieved peak yields with minimal additional benefit afterward (Figure 1). As such, the existing methods in the literature, such as two hours of hot water extraction and 72 h of ethanol extraction [17], are either insufficient for efficient saponin recovery or unnecessarily time-consuming.

Compared with the subsequent scCO_2_ extraction (Appendix A) that yielded saponins at (6.48 ± 0.36) mg OAE/g, subsequent 24 h hot water extraction (7.50 ± 0.25) mg OAE/g) and EtOH extraction (10.77 ± 0.51) mg OAE/g) produced significantly more saponins (*p* ≤ 0.05), among which, 70% EtOH recovered the highest total saponin amount (~16.26 mg OAE/g in scCO_2_-EtOH extraction, comparable to 16.44 mg OAE/g in hexane defatting–70% EtOH extraction and 17.31 mg OAE/g in hexane defatting–ultrasonic-assisted extraction). Therefore, while the scCO_2_ technique is not the most effective green tool for saponin extraction from *C. frondosa* viscera, a sequential approach integrating scCO_2_ with EtOH extraction emerges as a practicable alternative.

### 2.4. Comparison of Saponin Yields from Different Methods

The comparison of the performance of sequential scCO_2_ extraction and other sequential extraction methods on saponins derived from *C. frondosa* viscera is presented in Table 3 and Figure 2. Several saponin extraction methods, especially hexane extraction followed by hot water extraction (22.76% recovery efficiency), exhibited lower recovery efficiencies than the benchmark resulting from hexane defatting–ultrasonic-assisted extraction. In contrast, 70% EtOH extraction on hexane-defatted samples (16.44 ± 0.21 mg OAE/g) and scCO_2_ extraction followed by 70% EtOH extraction (16.26 ± 2.47 mg OAE/g) exhibited comparable saponin yields compared to hexane defatting–ultrasonic-assisted extraction (17.31 ± 0.60 mg OAE/g, *p* = 0.992 and 0.976 > 0.05). Additionally, 100% EtOH extraction on hexane-defatted viscera (10.19 ± 0.20 mg OAE/g), hexane extraction followed by reflux extraction (7.17 ± 0.78 mg OAE/g), and sequential scCO_2_ extraction (9.13 ± 1.30 mg OAE/g) yielded proportionate saponins (*p* > 0.05). Since the temperature, pressure, EtOH concentration, and extraction time of scCO_2_ extraction did not affect the sea cucumber saponin extraction yields, the relatively mild conditions (35 °C and 20 MPa or even the critical point of scCO_2_) can be used for better elution of saponins in a short period (30 min). Therefore, sequential scCO_2_ extraction outperforms certain conventional extraction methods (i.e., hexane extraction followed by 100% EtOH or reflux extraction) regarding efficiency and sustainability, offering a timesaving, environmentally friendly alternative for saponin isolation. While scCO_2_ alone yielded modest results (9.13 mg OAE/g), its integration with subsequent solvent extraction steps significantly enhanced performance. ScCO_2_ extraction followed by 24 h 70% EtOH extraction achieved 16.26 mg OAE/g, nearly matching the hexane defatting–ultrasonic-assisted method (17.31 mg OAE/g). Also, scCO_2_-coupled 24 h hot water extraction led to 12.99 mg OAE/g, with no statistical significance compared to 70% EtOH-participated ones.

Effective saponin extraction and analysis require the elimination of lipid interference, making pretreatment a crucial step in the process. The initial scCO_2_ extraction step served as a pretreatment by disrupting the structural integrity of the viscera matrix, thereby enhancing mass transfer efficiency during subsequent extraction and promoting saponin release. This mechanistic hypothesis aligns with prior research on scCO_2_ extraction of lipids from *C. frondosa* viscera, which documented significant microstructural alterations in the tissue matrix following lipid removal (i.e., pore formation and increased surface area) [21]. These structural modifications likely facilitated solvent penetration and saponin diffusion during downstream extraction. Comparison between conventional hexane defatting–hot water extraction and scCO_2_ extraction followed by hot water extraction indicates that the latter compensates for the relatively weak solvating power of water, leading to higher saponin yields even after one-hour extraction (5.57 ± 0.41 mg OAE/g, *p* = 0.0081 ≤ 0.05). While a three-day extraction with 70% ethanol on hexane-defatted samples yielded comparable levels (16.44 mg OAE/g), the scCO_2_-EtOH sequential extraction approach eliminates the use of toxic hexane, significantly reduces extraction time, and has been previously validated for superior fatty acid recovery [21].

### 2.5. Antioxidant Activities of Extracts

Given the reported antioxidant properties of sea cucumber saponins, such as their ability to scavenge DPPH radicals [18,19], the DPPH assay was conducted to assess whether different extraction methods influenced not only the yield but also the functional bioactivity of the resulting fractions from *C. frondosa* viscera (Table 4 and Figure 3). Previous studies have demonstrated that processing methods can significantly impact the antioxidant and physicochemical properties of sea cucumber products. For example, microwave-treated *Holothuria scabra* powder exhibited higher DPPH radical-scavenging activity compared to smoking and steaming treatments [48]. In this context, assessing antioxidant activity helps provide insight into the preservation or enhancement of bioactive quality through different extraction strategies.

The DPPH radical is a commonly used model system for evaluating the free radical-scavenging activity of samples based on their ability to transfer electrons or donate hydrogen. The results showed that the crude extracts have a limited ability to decrease the DPPH concentration, with a 2.41–19.47% radical-scavenging ability at 10 mg/mL under the test conditions, among which the lipids-rich scCO_2_ extracts (from original samples) and the hot water extracts (from samples defatted by scCO_2_ technique) showed the lowest antioxidant activity at (4.43 ± 1.00) and (6.57 ± 4.39)%, respectively. In contrast, the subsequent scCO_2_ extracts (from samples defatted by scCO_2_ technique) and the EtOH extracts (from samples defatted by scCO_2_ technique) exhibited comparable high antioxidant activity at (16.14 ± 1.98) and (17.12 ± 4.20)%. The scCO_2_ extraction and subsequent ethanol extraction following scCO_2_ lipid removal yielded more than twice the antioxidant activity compared to both hexane defatting–ultrasonic-assisted extraction and hot water extraction after scCO_2_ lipid removal. The strong antioxidant activity observed in the subsequent EtOH and scCO_2_ extracts can be attributed to the relatively mild extraction conditions (relatively low temperatures and an inert (oxygen-free) atmosphere of scCO_2_) and the high efficiency of these solvent systems in extracting saponins and other antioxidant compounds (e.g., carotenoids).

Although ultrasound generates cavitation bubbles that collapse violently, producing localized high temperatures (~5000 K) and pressures (up to 1800 atm) [49], then disrupting cells to release saponins, excessive energy input can damage their structure, such as glycosidic bond cleavage [50], reducing solubility and altering radical-scavenging capacity. A similar phenomenon in *Ganoderma lucidum* polysaccharides extracted by ultrasound and hot water was reported by Kang et al. (2019) [51].

**Table 4 marinedrugs-23-00272-t004:** The antioxidant activity of saponin-containing extracts from *C. frondosa* viscera obtained through different treatments (*n* = 3, mean ± SD).

Extract ^1^	Hexane Defatting–Ultrasonic-Assisted Extraction	ScCO_2_ Lipid Extraction	Subsequent scCO_2_ Saponin Extraction	70% EtOH Extraction Following scCO_2_ Lipid Extraction	Hot water Extraction Following scCO_2_ Lipid Extraction	Butylated Hydroxytoluene (BHT)
Average DPPH scavenging (%) ^2^	8.11 ± 1.16 ^b^	4.43 ± 1.00 ^b^	16.14 ± 1.98 ^a^	17.12 ± 4.20 ^a^	6.57 ± 4.39 ^b^	20.39 ± 0.68 ^a^ (tested)/23.13 (theoretical ^3^)
Antioxidant power (%) ^4^	39.75	21.74	79.14	83.92	32.23	\

^1^ The concentration of the extracts is 10 mg/mL (final conc.: 1.11 mg/mL), and the concentration of BHT is 100 μM (final conc.: 11.11 μM). The examined extracts were obtained through ultrasonic-assisted extraction from samples that were defatted using hexane, scCO_2_ extraction of native samples, subsequent scCO_2_, 24 h 70% EtOH, and hot water extraction, respectively, of samples that have undergone defatting via scCO_2_ extraction. ^2^ Average DPPH scavenging (%) values that do not share a letter are significantly different at α = 0.05 level by Tukey comparison. ^3^ The theoretical calculation follows the finding that BHT can scavenge 1.85 DPPH radicals per molecule [52]. ^4^ The antioxidant power is expressed as the fraction of the average DPPH scavenging % between extracts and BHT.

Furthermore, the results suggest that the initial stage of scCO_2_ extraction primarily removes a higher proportion of lipids enriched in polyunsaturated fatty acids (PUFAs), resulting in a lower relative concentration of potent antioxidants and consequently reducing the overall scavenging capacity. PUFAs are highly susceptible to lipid peroxidation, a chain reaction triggered by free radicals. In the presence of DPPH, PUFAs may undergo oxidation, consuming free radicals while generating secondary oxidation products (e.g., lipid hydroperoxides) that do not effectively quench DPPH [53]. The DPPH scavenging activity of the extracts from defatted samples further corroborated this finding, suggesting that lipid removal enhanced the antioxidant capacity of saponin-containing extracts.

Compared with the reference standard, BHT, the extracts from subsequent scCO_2_ extraction and EtOH extraction following scCO_2_ lipid extraction showed comparable antioxidant ability (*p* ≤ 0.05). This suggests that these natural extracts, when prepared at a concentration of 10 mg/mL, could serve as effective alternatives to BHT at a concentration of 100 μM.

### 2.6. Green and Efficient Valorization via Sequential Extraction

Beyond extraction efficiency and antioxidant activity, it is also important to consider the broader environmental implications, particularly in the context of green chemistry and future industrial scalability. Among all evaluated methods, scCO_2_ extraction followed by 24 h 70% EtOH extraction represents a favourable process with respect to extraction efficiency, environmental performance, and adherence to green chemistry principles. This sequential approach achieved high saponin yields and superior fatty acid recovery while relying only on low-toxicity solvents. In contrast to conventional solvent systems employing hazardous volatile organics like hexane, which contribute to photochemical smog and require strict hazardous waste management [54], this method significantly reduces worker exposure risks and minimizes the potential for environmental contamination.

When operated in a closed-loop system, scCO_2_ extraction virtually eliminates solvent emissions during operation. Furthermore, CO_2_ used in this method is recyclable and can be sourced from industrial by-products, contributing to sustainability. The process is also compatible with renewable energy for heating and pressurization, supporting broader environmental goals. As such, although the scCO_2_ technique involves higher energy inputs and therefore higher global warming potential, this trade-off is offset by the elimination of solvent hazards and improved recovery of high-value bioactive compounds. A life cycle assessment indicated that energy use for heating and solvent production is the primary contributor to environmental impacts in lab-scale extractions. However, these impacts can be substantially mitigated through process scale-up, the use of energy-efficient equipment, and the integration of renewable energy sources. For example, transitioning from small-scale extraction to a 30-litre system powered by a biomass boiler substantially lowered global warming potential and overall environmental burden [55].

Additionally, first-stage scCO_2_ extraction significantly contributes to the effectiveness of the sequential protocol by enabling the selective recovery of lipid fractions while enhancing the accessibility of saponin-rich components. Under optimized conditions, scCO_2_ extraction yielded high levels of fatty acids, particularly omega-3 fatty acids (yielding 16.30 ± 0.66 g of total fatty acids and 3.38 ± 0.20 g of EPA and DHA per 100 g of dried samples), and surpassed conventional organic solvent methods in both efficiency and selectivity [21]. As both fatty acids and saponins are key targets, removing lipids in the first step reduces interference and allows for more efficient and selective saponin extraction in the subsequent ethanol stage.

Considering its superior extraction performance, favourable antioxidant activity of the recovered fractions, and alignment with green chemistry principles, including safer solvent use, recyclability, and waste minimization, the sequential scCO_2_-EtOH extraction method represents a sustainable and scalable strategy for the valorization of sea cucumber processing waste.

## 3. Materials and Methods

### 3.1. Chemicals and Materials

Hot air-dried *C. frondosa* viscera (moisture content is around 6.45%) were provided by AKSO Marine Biotech Inc. (Hacketts Cove, NS, Canada). The viscera samples were ground into powder with sizes smaller than 18-mesh. The ground samples were packed in zipper bags and stored in the freezer for future use. All the chemicals and reagents with at least ACS reagent grade were purchased from Sigma-Aldrich (Oakville, ON, Canada), Thermo Fisher Scientific (Ottawa, ON, Canada), and VWR (Mississauga, ON, Canada).

### 3.2. Sequential ScCO_2_ Extraction of Lipids and Saponins

Hot air-dried, ground *C. frondosa* viscera (~1.5 g, Mettler Toledo AT261 Analytical Balance, Columbus, OH, USA) were subject to scCO_2_ extraction (SFE-110, Supercritical Fluid Technologies Inc., Newark, DE, USA) under the optimal conditions (75 °C and 45 MPa, with a 20 min static and a 30 min dynamic extraction, and an initial loaded co-solvent at 2:1 of 95% EtOH to feedstock mass ratio) for lipid extraction as previously described [21], obtaining the lipid-rich extracts (the 1st extracts). Defatted residual samples that remained in the scCO_2_ vessel were subjected to further extraction for saponin recovery. Without removing the residues, a certain amount of EtOH (concentration at 50–100% *v*/*v*, 0–7.5 mL) was pumped into the extraction vessel. Once the temperature (35–75 °C) and pressure (20–50 MPa) reached the set point, the static stage (0–30 min) was recorded. After the pre-determined static soaking, the static/dynamic valve was opened, and the restrictor valve was slightly adjusted to ensure scCO_2_ was continuously drained through the vessel at approximately 10 mL/min. The co-solvent pump introduced the EtOH into the system at 0.5 mL/min during the entire dynamic extraction time (30–90 min). The continuous feeding of EtOH co-solvent was determined in the preliminary single-factor experiments. After completing the two-step extraction, the system was vented until the pressure inside the vessel dropped to atmospheric pressure and the temperature reached room temperature. The vessel was opened to remove the residues and then reconnected to the system for washing. Then, 95% EtOH was purged using the co-solvent pump to rinse the line and the vessel to collect the remaining extract. The extract obtained from system washing was combined with the extracts obtained in the subsequent extraction, regarding the 2nd extracts (Figure 4). All the extracts were dried under reduced pressure using a rotary evaporator and stored in the fridge for further purification. The single-factor experiments were performed in duplicate, while the definitive screening experiments were generated by Minitab 21.1.0 (Minitab Inc., State College, PA, USA).

### 3.3. Conventional Extraction of Saponins

To evaluate the scCO_2_-assisted sequential extraction, documented conventional extraction methods (Figure 5) were performed. Pre-weighed ground, hot air-dried *C. frondosa* viscera was defatted by soaking in hexane three times (~30 min), with the final soak left overnight. The conventionally defatted samples after vacuum oven drying (105 °C overnight, Across International, Livingston, NJ, USA) were subjected to the following traditional methods for saponin extraction.

#### 3.3.1. EtOH Extraction

The extraction method was adapted from Fagbohun et al. (2024) [17]. Native and defatted *C. frondosa* viscera (~5 g) were combined with 100 mL of absolute or 70% EtOH, respectively, and placed in a shaker (Benchtop Shaking Incubator, Corning, NY, USA) at 200 rpm for 72 h. The different extraction solvent concentrations were set to examine their affinity to saponins. Each extraction was performed in three replicate samples. To explore the effects of lipid interference, the resulting extracts were not subject to purification.

#### 3.3.2. Reflux Extraction

A reflux extraction of saponins using the 60% EtOH method developed by Gao et al. (2014) [25] was adopted. Native and defatted *C. frondosa* viscera (~1.5 g) were subjected to 20 mL of 60% EtOH in a reflux setup at 85 °C for one hour. The extraction was repeated three times. The combined extracts were concentrated using a rotary evaporator under reduced pressure. Each extraction was performed in three replicate samples. The dried extracts were further purified before analysis.

#### 3.3.3. Hot Water Extraction

Defatted *C. frondosa* viscera (~5 g) were soaked in 100 mL of distilled water and placed in a water bath (Precision 280 series water bath, Thermo Fisher Scientific, Waltham, MA, USA) at 80 °C for two hours [17]. The extraction was performed in three replicate samples.

#### 3.3.4. Ultrasonic-Assisted Extraction

Defatted *C. frondosa* viscera (~5 g) were soaked in 100 mL of 80% EtOH and placed in an ultrasonic water bath (50–60 Hz, VWR Aquasonic 750D Ultrasonic Cleaner, VWR, Radnor, PA, USA), with a temperature set at 60 °C for 52 min [17]. The extraction was performed in three replicate samples.

### 3.4. ScCO_2_ Extraction Followed by Conventional Extraction

Hot air-dried, ground *C. frondosa* viscera (~1.5 g) were subject to scCO_2_ extraction under the optimal lipid extraction conditions: 75 °C and 45 MPa, with a 20 min static and a 30 min dynamic extraction, and co-solvent (95% EtOH) to feedstock mass ratio of 2:1 [21]. The residues were removed from the scCO_2_ system and placed in centrifuge tubes filled with 50 mL of distilled water for hot water extraction or 70% EtOH for EtOH extraction. The operating procedure was the same as described above. To determine the extraction yield plateau and shorter durations with the maximum yields, the saponin contents in the extracts were measured in half an hour, one hour, five hours, 24 h, 48 h, and 72 h (Figure 6). Each extraction was performed in three replicate samples.

### 3.5. Purification

The purification procedure was adapted from Gao et al. (2014) [25]. The dried extracts were suspended in 25 mL of distilled water and partitioned against 20 mL of n-hexane thrice. The biphasic mixtures were centrifuged (Eppendorf Centrifuge 5804, Eppendorf, Hamburg, Germany) at 10,000 rpm for 10 min, followed by removing the upper n-hexane layer using Pasteur pipettes. The aqueous layer was partitioned using 15 mL of water pre-saturated n-butanol three times. Then, the combined n-butanol fraction was partitioned against 30 mL of n-butanol pre-saturated water. The collected n-butanol fraction was dried under reduced pressure for colorimetric analysis in a UV-Vis spectrophotometer (DR6000 Benchtop Spectrophotometer, Hach, Loveland, CO, USA).

### 3.6. Saponin Yield Determination

#### 3.6.1. Standard Solution and Stock Solution Preparation

To prepare the standard oleanolic acid solution, 9.96 mg oleanolic acid was weighed and dissolved in MeOH, followed by transferring to a 50 mL volumetric flask. The 5% vanillin–acetic acid solution was prepared by dissolving 0.50 g vanillin in glacial acetic acid and then adjusting the volume in a 10 mL volumetric flask.

#### 3.6.2. Standard Curve Preparation

The standard oleanolic solution was drawn at 0 (blank), 0.1, 0.2, 0.3, 0.4, 0.5, and 0.6 mL in 12 mL test tubes with lids. The solutions were concentrated under the nitrogen stream and then reconstituted using 0.2 mL of 5% vanillin–acetic acid solution and 0.8 mL of perchloric acid. The well-mixed solutions were placed on a reactor block at 60 °C for 15 min and then cooled in the ice water bath. Before colorimetric analysis, 5 mL of acetic acid was added, and the mixtures were set at room temperature for 10 min. The solutions were subjected to a UV-Vis spectrophotometer at 400–650 nm to find the wavelength demonstrating maximum absorbance for oleanolic acid. The wavelength with the maximum absorbance was found at 546 nm. Each tube was measured at 546 nm six times to obtain the average absorbance. A standard curve (Appendix A) was plotted using the amount of OAE as the *X*-axis and the absorbance as the *Y*-axis.

#### 3.6.3. Total Saponin Content Determination

The dried fractions (from extraction or purification) were dissolved in 25 mL of 60% MeOH. Then 0.25 mL of the extract-rich solution was pipetted into the 12 mL test tubes with lids. The solutions were concentrated under the nitrogen stream and then reconstituted using 0.2 mL of 5% vanillin–acetic acid solution and 0.8 mL of perchloric acid. The well-mixed solutions were placed on a reactor block at 60 °C for 15 min and then cooled in the ice water bath. Prior to colorimetric analysis at 546 nm, 5 mL of acetic acid was added, and the mixtures were set at room temperature for 10 min. Each tube was measured six times, and the OAE in the samples (OAE mg/g of samples on a dry weight basis) was calculated based on the standard curve (Y = 10.874X − 0.0426).

### 3.7. Antioxidant Activity Test

The DPPH free radical-scavenging activity of sea cucumber saponin extracts was assayed using the DPPH method according to the modified procedure from the previous work [56]. The 0.10 mM DPPH solution was prepared by dissolving 1.97 mg of DPPH in 70% MeOH in a 50 mL amber volumetric flask. To determine the DPPH radicals scavenged, 0.25 mL of the extracts in 70% MeOH (10 mg/mL) was mixed with 2 mL of 0.10 mM DPPH solution in a 12 mL test tube. A control was prepared by mixing 0.25 mL 70% MeOH with 2 mL of DPPH solution. A blank with 2.25 mL of 70% MeOH was set to zero the absorbance. Samples were vortexed for 15 s and held at room temperature in the dark (wrapped with the foil) for 30 min. After incubation, the samples and the blank were measured for their absorbance six times at 517 nm using a UV-Vis spectrophotometer. Then 100 µM of BHT was used as reference for comparison. The percentage of DPPH radical-scavenging (% inhibition) was calculated:(1)$$\%\;DPPH\;inhibition=\left[1-\left(\frac{Absorbance\;of\;a\;sample\;at\;517\;nm}{Absorbance\;of\;the\;control\;at\;517\;nm}\right)\right]\times 100\%$$

### 3.8. Statistical Analysis

The results are presented as mean ± standard deviation (mean ± SD). Statistical analyses were performed using *T*-tests or one-way ANOVA followed by Tukey comparison, with differences considered significant at *p*  ≤  0.05.

To evaluate the impacts of the proposed process variables on the scCO_2_ extraction of sea cucumber saponins, preliminary single-factor experiments were conducted to establish a fundamental understanding. A definitive screening design, a three-level experiment design, was then implemented to investigate six potential variables simultaneously: temperature, pressure, static extraction time, dynamic extraction time, co-solvent amount in the static stage, and co-solvent concentration. Stepwise analysis (α = 0.05 or 0.15 for both entry and removal) was applied to identify significant linear, interaction, or quadratic terms. If any terms met the criteria for model entry, a response surface design would be conducted around those variables to determine optimized extraction conditions. However, as no terms were retained in the model, follow-up single-factor experiments were performed to confirm the observed trends and assess reproducibility under selected conditions.

## 4. Conclusions

This study successfully demonstrated that the sequential use of scCO_2_ extraction followed by hot water extraction or EtOH extraction offers an effective and green method for extracting saponins from *C. frondosa* viscera residues. The sequential scCO_2_-EtOH extraction method yielded 16.26 mg OAE/g, comparable to the benchmark conventional hexane defatted–ultrasonic-assisted extraction (17.31 mg OAE/g) but without intensive energy inputs and toxic organic solvent use. This streamlined process enhances operational efficiency and sustainability, making it suitable for industrial applications.

Regarding sequential scCO_2_ extraction, the suggested scCO_2_ conditions in the second step (35 °C, 20 MPa, 30 min dynamic extraction with continuous feeding of 75% EtOH at 0.5 mL/min) achieved a relatively average recovery (9.13 mg OAE/g) while maintaining higher energy efficiency, mild operating conditions, and less solvent intensity. The results indicated that varying process variables of the scCO_2_ technique did not help the recovery of saponins from *C. frondosa* viscera residues, and a mild supercritical condition with continued co-solvent feeding can flush out a moderate amount of saponins. The absence of significant differences across the tested ranges demonstrates the robustness of the process to minor fluctuations. Given the non-negligible variations inherent to supercritical systems, it is a highly desirable attribute for large-scale industrial applications where maintaining consistent scCO_2_ extraction performance is essential.

Defatting and purification steps were found to be crucial in improving saponin recovery, effectively reducing lipid interference and enhancing solvent penetration. The first scCO_2_ extraction for lipid removal, acting like a pretreatment for subsequent saponin extraction, improved the performance of the subsequent extraction by disrupting the structural integrity of the biological matrix, promoting better mass transfer and saponin elution. The antioxidant activities of scCO_2_-EtOH extracts (17.12% DPPH scavenging) and sequential scCO_2_ extracts (16.14%) were comparable to the synthetic antioxidant BHT, further supporting the bioactivity potential of these extracts.

Overall, this study, along with the previous study [21], highlights the potential of combining optimized scCO_2_ and EtOH extraction as a viable method for the sequential recovery of fatty acids and saponins from sea cucumber by-products. This sequential extraction approach offers a scalable, solvent-conscious, and resource-efficient solution for the valorization of marine processing residues.

## Figures and Tables

**Figure 1 marinedrugs-23-00272-f001:**
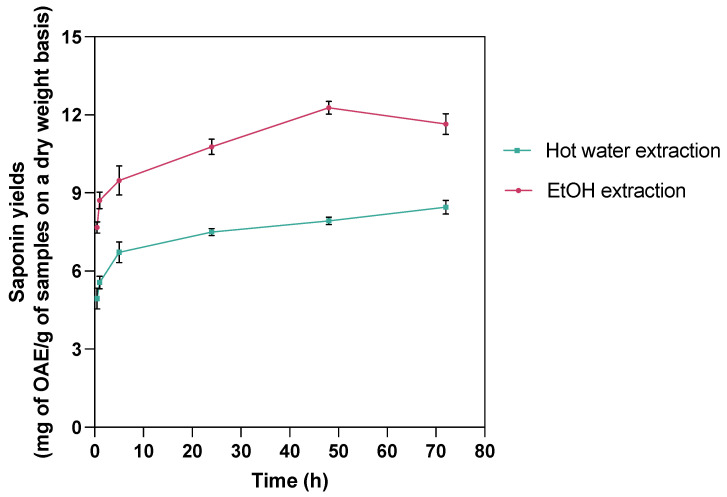
Time-dependent changes in saponin yields from hot water and 70% EtOH extractions following scCO_2_ lipid extraction at various time intervals (0.5, 1, 5, 24, 48, 72 h). Results demonstrate variation in extraction efficiency over time, highlighting optimal time for maximum saponin recovery for each extraction method.

**Figure 2 marinedrugs-23-00272-f002:**
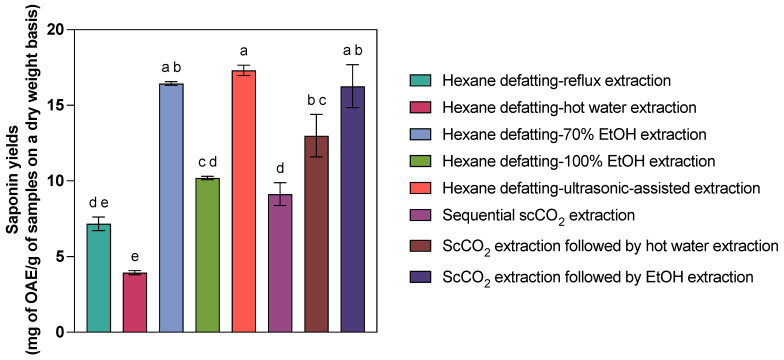
Comparison of saponin yields (mg OAE/g of samples on a dry weight basis) from *C. frondosa* viscera via different sequential extraction methods. Bars labelled with different letters indicate statistically significant differences among extraction methods based on Tukey’s HSD test at α = 0.05 level.

**Figure 3 marinedrugs-23-00272-f003:**
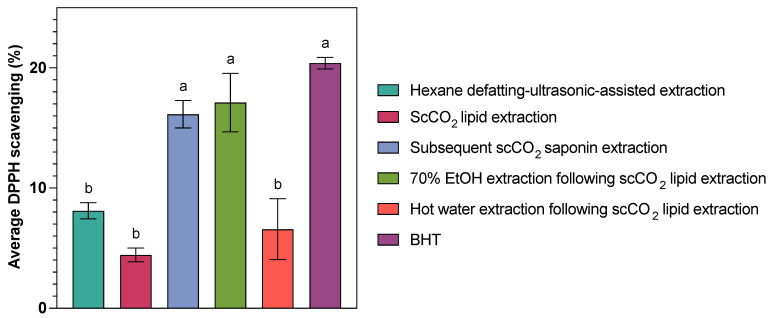
Comparison of average DPPH scavenging (%) of saponin extracts from *C. frondosa* viscera obtained from different methods. Bars labelled with different letters indicate statistically significant differences among extraction methods based on Tukey’s HSD test at α = 0.05 level.

**Figure 4 marinedrugs-23-00272-f004:**
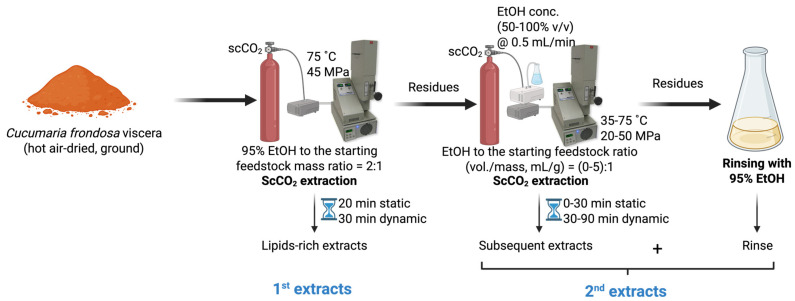
The scheme of sequential scCO_2_ extraction of *C. frondosa* viscera.

**Figure 5 marinedrugs-23-00272-f005:**
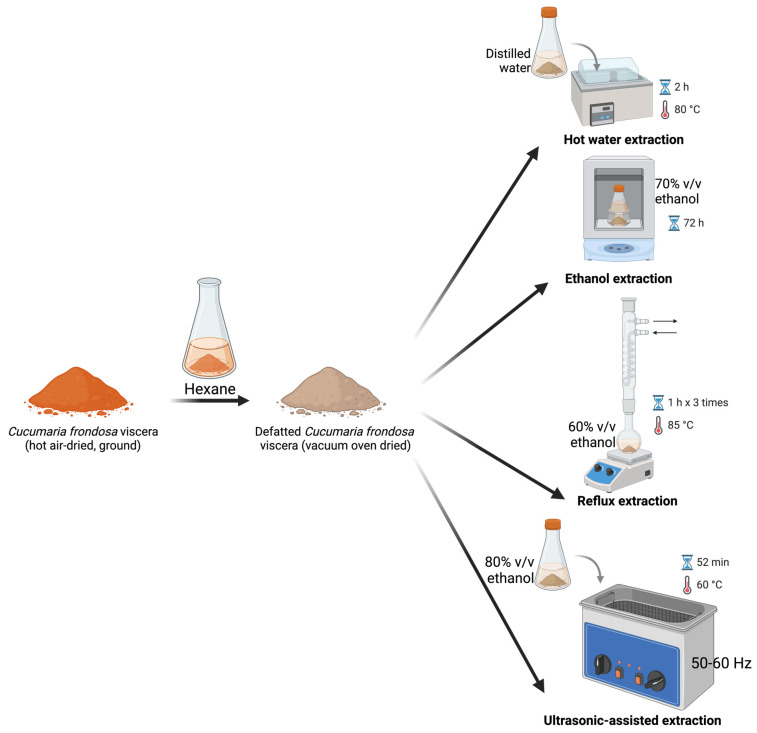
Conventional extraction of saponins from hexane-defatted *C. frondosa* viscera.

**Figure 6 marinedrugs-23-00272-f006:**
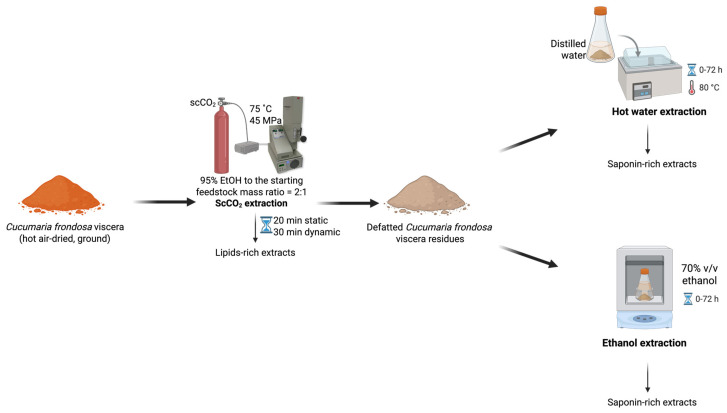
ScCO_2_ extraction followed by hot water extraction or EtOH extraction on *C. frondosa* viscera.

**Table 1 marinedrugs-23-00272-t001:** Comparison of saponin yields (mg OAE/g of samples on dry weight basis, *n* = 3, mean ± SD) resulting from reflux and EtOH extraction on native and defatted *C. frondosa* viscera.

Treatment *	Reflux-Native	Reflux-Defatted	70% EtOH-Native	70% EtOH-Defatted	100% EtOH-Native	100% EtOH-Defatted
Yields (mg OAE/g)	4.18 ± 0.82	6.71 ± 0.73	36.34 ± 0.64	16.44 ± 0.21	38.22 ± 0.83	10.19 ± 0.20

* The reflux extraction was followed by purification (liquid–liquid partitioning using hexane and butanol) before colorimetric analysis, while EtOH extracts were subject to colorimetric analysis directly. Native means original biomass as initial feedstock, while defatted refers to hexane-defatted starting materials.

**Table 2 marinedrugs-23-00272-t002:** Comparison of saponin extraction yields (mg of OAE/g of samples on dry weight basis, *n* = 3, mean ± SD) from hot water and EtOH extraction of scCO_2_-pretreated, lipid-removed *C. frondosa* viscera over time. Means in the same row with different letters are significantly different at α = 0.05 level by Tukey comparison.

	Time (h)	0.5	1	5	24	48	72
Treatment							
Hot water extraction		4.94 ± 0.69 ^a^	5.57 ± 0.41 ^a,b^	6.72 ± 0.69 ^b,c^	7.50 ± 0.25 ^c,d^	7.93 ± 0.24 ^c,d^	8.45 ± 0.46 ^d^
EtOH extraction		7.68 ± 0.38 ^a^	8.72 ± 0.55 ^a,b^	9.47 ± 0.97 ^b,c^	10.77 ± 0.51 ^c,d^	12.27 ± 0.43 ^d^	11.65 ± 0.69 ^d^

**Table 3 marinedrugs-23-00272-t003:** Comparison of *C. frondosa* viscera saponin yields (mg OAE/g of samples on dry weight basis, mean ± SD, *n* = 3) resulting from different sequential extraction methods. Means in the same column with different letters are significantly different at α = 0.05 level by Tukey comparison.

Method ^1^	Yields (mg OAE/g) ^2^	Recovery Efficiency ^3^
Hexane defatting–70% EtOH extraction	16.44 ± 0.21 ^a,b^	96.13%
Hexane defatting–100% EtOH extraction	10.19 ± 0.20 ^c,d^	58.87%
Hexane defatting–hot water extraction	3.94 ± 0.23 ^e^	22.76%
Hexane defatting–reflux extraction	7.17 ± 0.78 ^d,e^	41.42%
Hexane defatting–ultrasonic-assisted extraction	17.31 ± 0.60 ^a^	/
Sequential scCO_2_ extraction	9.13 ± 1.30 ^d^	52.74%
ScCO_2_ extraction followed by hot water extraction	12.99 ± 2.43 ^b,c^	75.04%
ScCO_2_ extraction followed by EtOH extraction	16.26 ± 2.47 ^a,b^	93.93%

^1^ Hexane-defatted samples were subsequently extracted by 72 h 70% EtOH extraction, 72 h 100% EtOH extraction, two-hour hot water extraction, reflux extraction, and ultrasonic-assisted extraction, while the rest were scCO_2_ extraction followed by scCO_2_ extraction, 24 h hot water extraction, and 24 h EtOH extraction. ^2^ The yields include the saponins presented in the lipid-rich extracts, rinse, and extracts from the subsequent extraction. ^3^ The extraction yields of ultrasonic-assisted extraction were used as a reference for efficiency calculation.

## Data Availability

The original contributions presented in the study are included in the article/Appendix A; further inquiries can be directed to the corresponding author.

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
