# Peer review of "Sequential Extraction of Bioactive Saponins from Cucumaria frondosa Viscera: Supercritical CO2–Ethanol Synergy for Enhanced Yields and Antioxidant Performance"

_marinedrugs, 2025, doi:10.3390/md23070272_

Round 1
Reviewer 1 Report (Previous Reviewer 1)
Comments and Suggestions for Authors
The authors have revised their previous question. However,some references are too old. It is recommended to cite those within the last five years.
Author Response
Thank you for highlighting the need to modernize our reference list. We have carefully reviewed every citation and replaced or supplemented all general background references with up-to-date literature. We have, however, retained a small number of classical papers, most notably the first isolation and structural elucidation reports of Cucumaria frondosa saponin fractions. These works provide essential historical context.
Reviewer 2 Report (Previous Reviewer 3)
Comments and Suggestions for Authors
Thank you for your clear and thoughtful replies to my comments. I believe the manuscript has been improved greatly with the new text and figures.
Author Response
Thank you for your kind words and for taking the time to review our revisions. Your constructive feedback was invaluable in shaping these improvements, and I sincerely appreciate your support.
This manuscript is a resubmission of an earlier submission. The following is a list of the peer review reports and author responses from that submission.
Round 1
Reviewer 1 Report
Comments and Suggestions for Authors
The article titled “Eco-Efficient Extraction of Bioactive Saponins from Cucumaria frondosa Viscera: Supercritical COâ‚‚-Ethanol Synergy for Enhanced Yield and Antioxidant Performance”. In this study, a green extraction method based on the synergistic effect of supercritical COâ‚‚ and ethanol was proposed for the efficient extraction of saponins from sea cucumber (Cucumaria frondosa) viscera and compared with traditional methods such as ultrasound-assisted extraction. The research provides new ideas for the high-value utilization of marine by-products, which is in line with the current needs of green chemistry and sustainable development. However, some small details need to be revised. Therefore, I recommend it to be accepted after a minor revision.
The specific content is as follows:
- In the introduction, the review of the biological activity and extraction methods of saponins is relatively detailed, but some discussion on the advantages and potential of sea cucumber viscera as an extraction raw material can be appropriately added to highlight the background and significance of the research.
- The extraction method of supercritical COâ‚‚ combined with ethanol performed well in terms of saponin yield and antioxidant activity, but further explanation is needed as to why the extraction efficiency of supercritical COâ‚‚ alone was low (9.13 mg OAE/g), while the subsequent ethanol extraction significantly improved it (16.26 mg OAE/g).
- The antioxidant activity data (such as DPPH clearance) need to be supplemented with statistical comparisons with BHT (such as p-values) to clarify the actual significance of their biological activity.
- The resolution of the charts in Figures 2 and 3 needs to be improved, and some labels (such as the horizontal and vertical axes) should be clearer.
- The reference format should be standardized according to the journal requirements.
- It is recommended that Figure A1 and Table A1 be placed in the Supplementary material and uploaded to the system separately.
Reviewer 2 Report
Comments and Suggestions for Authors
Manuscript entitled “Eco-Efficient Extraction of Bioactive Saponins from Cucumaria frondosa Viscera: Supercritical CO2-Ethanol Synergy for Enhanced Yield and Antioxidant Performance”
After careful evaluation, I conclude that this manuscript does not meet the publication standards of the journal. Therefore, I recommend rejecting this submission.
- The literature review, while referencing pertinent publications, presents limited critical analysis and synthesis. It inadequately delineates how this study diverges from or builds upon prior research, thereby obscuring its original scholarly significance.It is recommended to further expand and refine the literature review section by strengthening the systematic synthesis and critical analysis of existing research findings, thereby better underscoring the unique contributions of this study within the field.
- The experimental design appears oversimplified, and the supporting data/graphical representations lack sufficient depth. This methodological limitation undermines the analytical rigor required to validate the research claims, necessitating expanded datasets and enhanced visual documentation.It is recommended to incorporateorthogonal experimental design (OED) or multivariate response optimization (MRO) to systematically investigate variable interactions and enhance the robustness of the experimental framework.
- The evaluation of extraction efficiency is incomplete. The discussion of scCOâ‚‚ extraction performance focuses solely on saponin yield, while neglecting critical analyses of co-extracted constituents (e.g., lipid removal efficiency) and their impacts on saponin purity. It is essential to supplement these data and discussions to provide a holistic assessment of scCOâ‚‚ extraction's advantages and operational effectiveness.
- The current comparison of extraction methods focuses narrowly on saponin yield and antioxidant activity, omitting critical practical considerations such as extraction costs, time efficiency, equipment specifications, and environmental impacts. This limited scope may lead to biased conclusions regarding method applicability. It is imperative to conduct a multidimensional evaluation integrating techno-economic analysis and sustainability metrics to provide actionable guidance for industrial implementation.
- The exclusive reliance on spectrophotometric methods (colorimetric assays) for saponin quantification introduces significant analytical uncertainty due to potential interference from co-extracted compounds. The absence of chromatographic validation (e.g., HPLC, LC-MS) undermines the methodological rigor of the findings. It is imperative to implement hyphenated chromatography techniques to verify spectrophotometric data accuracy, particularly for comparative analyses of scCOâ‚‚-EtOH extracts versus ultrasound-assisted extracts.
- Certain unit symbols (e.g., "mg OAE/g") and abbreviations (e.g., BHT) require explicit definition upon their first occurrence in the text.
- Throughout the text, the statistical significance notation should be formatted asp ≤ 0.05(with italicized "p") .
- The quality of English needs improving. It is noted that your manuscript needs careful editing by someone with expertise in technical English editing paying particular attention to English grammar, spelling, and sentence structure so that the goals and results of the study are clear to the reader.
- Please check carefully the format of references.
Reviewer 3 Report
Comments and Suggestions for Authors
The manuscript describes the analysis of a new extraction regime for triterpenoid saponins from sea cucumbers. It is an interesting look into bioprocessing of marine samples for bioactive compounds.
1. It would be useful to have Figures 4-6 moved to places earlier in the text, as I find the visual depictions of the extraction processes useful when trying to understand the results presented. Perhaps having them within Section 2 (and then citing Materials and Methods) could help readers understand the process earlier and help make sense of your results. If not possible to move, I would suggest adding some notes in the areas of lines 64-66 and 100-105 indicating to the reader that there are pictographs available later in the manuscript.
2. There seems to be a focus on the spectrophotometric analysis of saponin extraction methods in this manuscript; however, I would ask why you did not subject your samples to mass spectrometry analysis in order to validate your spec methods for this study? LC-MS analysis of triterpenoids is well-established and inclusion of spectra would help support your claims of interference/contamination of unwanted compounds under certain extraction regimes. Is MS analysis of your samples still possible? I think their inclusion, along with spectra of known saponin standards, would greatly support your claims in this manuscript for readers not familiar with these processes, as much of the data presented is in numerical format.
3. Regarding Figure 2 and Tables 3 & 4, it would be helpful to have a legend denoting which conditions you are referring to when using the labels cd, ab, bc, etc in figure and tables. I believe you outline the conditions in lines 260-283 but I do not see any indication of what these labels mean, other than by inferring them from Figure 2.
4. Also, regarding Figure 2, it could be possible to make this legend I am asking for in the figure label by moving the long text on the x axis to the figure legend, associating it with the appropriate labels there, and then move the labels from on top of the data bar to the x axis. This may also make Figure 2 easier to interpret. This is also suggested for Figure 3 as well.
5. For section 3.5, it is not readily apparent why you chose to test your fractions for antioxidant activity, as I believe antioxidative activity are only briefly mentioned on line 42. A line or two about the antioxidant activity of saponins and how it might relate to your extraction methods could be useful to help remind the reader of why this assay was carried out.